# Differences According to Sex in the Relationship between Social Participation and Well-Being: A Network Analysis

**DOI:** 10.3390/ijerph192013135

**Published:** 2022-10-12

**Authors:** Di Zhao, Guopeng Li, Miao Zhou, Qing Wang, Yiming Gao, Xiangyu Zhao, Xinting Zhang, Ping Li

**Affiliations:** 1Department of Health Psychology, School of Nursing and Rehabilitation, Shandong University, Jinan 250012, China; 2Department of Pediatrics, Qilu Hospital of Shandong University, Jinan 250012, China

**Keywords:** social participation, well-being, sex differences, network analysis

## Abstract

This study aimed to explore the effects of different types of social participation on the components of well-being, as well as the differences according to sex in the relationship between social participation and well-being. This was a cross-sectional community study. Well-being was measured using the positive emotion, engagement, relationships, meaning, and accomplishment (PERMA) profile. Social participation included activities of daily life, sports and entertainment activities, and social service activities. The independent association between each type of social participation and the components of well-being was examined by using a network analysis in both males and females. Of all 1276 participants surveyed, 60% were females. The results of the network analysis showed that positive emotion–housework (0.263), positive emotion–games (0.102), engagement–housework (0.107), engagement–work (0.054), and meaning–socializing (0.085) had unique connections in males; the average predictability in the network was 0.417. For females, positive emotion–physical activity (0.102), engagement–associations or societies (0.071), relationships–physical (0.090), relationships–socializing (0.092), and relationships–volunteering activities (0.133) had significant connections; the average predictability in the network was 0.358. Different types of social participation may have different effects on the components of well-being. Furthermore, differences according to sex in the relationship between social participation and well-being should be considered when formulating interventions to improve well-being.

## 1. Introduction

With the development of positive psychology, many researchers have considered well-being as their central research topic. Well-being is a measure of how well life goes for the person who lives it [1], which includes emotional stability, engagement, meaning, optimism, positive emotion, and positive relationships [2]. Psychological studies of well-being can be divided into hedonic and eudaimonic paradigms [3]. The hedonic view holds that well-being is a subjective state of happiness while the eudaimonic view believes that it is a person’s qualities and optimal functioning [4]. Seligman’s well-being theory integrated the hedonic and eudaimonic views of well-being and proposed the PERMA model [5], in which well-being consists of positive emotion (subjective experience of happiness in the past, present, and future), engagement (which reflects the use of the force of character and an individual’s talents and capacity), relationships (degree of creativity and altruism in social relationships), meaning (which indicates feeling like part of a greater purpose), and accomplishment (joy derived from striving for success or victory as a way to self-realization). Studies have indicated that well-being is associated with numerous health benefits such as a lower inflammatory response, cardiovascular risk, and morbidity [6,7,8]. Given the importance of well-being to human health, it is necessary to discover an effective approach to improve well-being.

Social participation is fundamental to human involvement in life situations and is an important modifiable health determinant [9]. It was defined as participation in formal or informal social activities, including paid or unpaid jobs, volunteering, and social groups, that provide interaction with others in the community [10]. Studies have pointed out that social participation could reduce individuals’ negative emotional experiences (e.g., depressive symptoms) [11] and enhance their well-being [12,13]. A study of 314 healthy adults demonstrated that individuals involved in volunteering or paid work reported higher levels of well-being [14]; social and community-related activities could promote better emotional health [15]. However, social activities are so varied that it is difficult to cover them comprehensively. Based on the framework of Levasseur and colleagues, there are six main levels of social activities: (1) doing an activity in preparation for connecting with others; (2) being with others; (3) interacting with others without doing a specific activity with them; (4) engaging in an activity with others; (5) helping others; and (6) contributing to society [10]. Therefore, depending on the goal of these social participation activities, we included 11 activities in this study: housework, transportation, physical activity, sports, arts and culture, games, socializing, work, donation, volunteering, and associations or social activities. These activities mainly referred to the three aspects of activities of daily life, sports and entertainment activities, and social service activities.

Studies have found that males and females have different levels of well-being and social participation. It is widely acknowledged that there are differences according to sex in social construction [16,17]. The social construct is also reflected in the differences in social participation [18], whereby females tend to be more engaged in household activities and in helping others than males, who appear to be more engaged in activities outside the home such as sports, socializing, and associational memberships [19,20]. Another study showed that females were more likely to engage in physical activity than males [15]. Research on the differences between males and females in social participation has not yielded conclusive results. Notably, sex is also an important determinant of health, including mental health [16]. Studies have reported that females had a significantly lower well-being than males, especially with regard to life satisfaction, personal growth, and positive emotions [21,22]. Conversely, another study indicated that females scored higher on well-being than males, mainly in terms of good relationships and personal growth [23]. Nevertheless, some studies have reported no differences in well-being according to sex [20,24]. This heterogeneity may be due to the diversity of the definitions of well-being and the measurement tools. Seligman’s well-being theory is generally accepted as covering the connotation of well-being [5]. Therefore, we used the PERMA Profiler tool to investigate individuals’ well-being. Based on the foregoing, we posited that males and females might have different types of social participation and levels of well-being.

In addition, studies indicated that well-being was more significantly related to activities at home for females compared to males; for males, performing activities with others (such as sports and cultural events) may contribute to higher levels of well-being [22,25]. A study showed a greater the effect of social interaction on well-being for females than for males [26]. However, there is still a lack of information on the relationship between different types of social participation and components of well-being based on sex. Considering the various types of social participation and components of well-being, it is essential to identify the complex associations between these variables. A network analysis is a sophisticated statistical method that can reveal complex associations between variables while controlling for their associations with all other variables (i.e., conditional dependence relationships) [27]. Furthermore, as a type of network analysis, a mixed graphical model (MGM) can process different types of data simultaneously (including categorical and continuous variables) [28].

Given the importance of social participation to well-being, this study probed the link between different types of social participation and the components of well-being in healthy adults, as well as examined whether there was a difference between them based on sex. From a theoretical perspective, we proposed the following hypotheses: (1) Social participation has a positive impact on an individual’s well-being; (2) different types of social participation have different effects on well-being and its components; and (3) there are differences according to sex between social participation and well-being.

## 2. Materials and Methods

### 2.1. Study Design and Sample

This was a cross-sectional community study. A convenient sample of participants was recruited using posters on community websites in Shandong Province, China, from October 2021 to January 2022. Brief information on the study, the participant criteria, and the contact information for the study investigator were provided on the study posters. Participants were included if they: (a) were between the ages of 18 and 65 years; (b) could read and understand Chinese; and (c) were willing to participate in the study. Those with any somatic disease condition that affected physical activity were excluded (such as fractures, knee injuries, and osteoarthritis). This study was approved by the authors’ institutional ethics committee (2021-R-048) and all participants provided informed consent.

Potential participants obtained questionnaires by scanning a QR code on the poster. A survey note accompanying the questionnaire informed participants of their anonymity, ensured voluntary participation in the study, and provided instructions on how to complete the questionnaire. We also stated the length of time it would take to complete the survey (5~10 min). The survey could only be submitted when participants completed all questions; therefore, there were no missing data. Participants completed the questionnaire via an online link and submitted it directly to the network questionnaire platform. To ensure the quality of the survey, only one questionnaire per IP address was accepted. A final total of 1276 participants were included in the analysis.

### 2.2. Measures

#### 2.2.1. Demographic Information

Participants provided sociodemographic information such as age, sex, marital status, education, and economic condition. Marital status (married and other (single, widowed, and divorced)), education (senior high school or below, college, and university or above), and economic condition (insufficient, sufficient for essentials, and more than sufficient) were the categorical variables.

#### 2.2.2. Social Participation

Participation in social activities was assessed using a researcher-designed questionnaire according to Levasseur’s framework [10] that mainly included activities of daily life (housework and transportation), sports and entertainment activities (physical activity, sports, arts and culture, games, and socializing), and social service activities (work, donation, volunteering, and associations or social activities) (Table 1). These activities are common types of individual activities and fit within the theoretical framework. Participants who reported attending any activity at least once per week were assigned a score of 1; those who attended less frequently or not at all were assigned a score of 0.

#### 2.2.3. Well-Being

The PERMA Profiler was used to evaluate individuals’ well-being levels [29]. The PERMA Profiler includes 23 items (15 PERMA items and 8 filler items). In this study, we used these 15 items to assess the five components of well-being (positive emotion, engagement, relationships, meaning, and accomplishment). Responses were scored on a Likert scale ranging from 0 to 10 (0 = not at all, 10 = completely; 0 = never, 10 = always; or 0 = terrible, 10 = excellent), with higher scores indicating a greater presence of the investigated component. This tool demonstrated good internal consistency (Cronbach’s alpha = 0.949).

#### 2.2.4. Analysis

All analyses were performed using IBM SPSS (version 22.0; SPSS, Inc., Chicago, IL, USA), R 4.0.2 (R Foundation for Statistical Computing, Vienna, Austria), and a variety of R packages. The continuous variables were presented as the mean (M) and standard deviation (SD). The categorical variables were reported as frequencies and percentages. The test level was 2-sided; *p* < 0.05 was considered statistically significant.

A mixed graphical model (MGM) was used to estimate network structure [28] among socio-demographic variables, well-being, and social participation in both male and female groups. The MGM can reveal relationships that remain hidden or underestimated when using univariate methods. In the network structure, variables are represented by nodes (including categorical and continuous variables) and the connections between variables are represented by edges. Two nodes are connected by an edge only if their relationship cannot be explained by any other node in the model or if their partial correlation coefficient is not equal to 0. Thus, it eliminates spurious relationships between variables and can potentially reveal new relationships adjusted for all other variables. The MGM estimates the regression coefficients representing the edge weights through nodewise regression. To estimate the network structure, we used a pairwise model (interaction order *k* = 2) and extended Bayesian information criterion (EBIC) to select the LASSO regularization parameter, which was applied and set at 0.5 to estimate the network structure. The thickness of each edge in the visualized network indicated the strength of the connection, with thicker edges indicating a stronger relationship [30].

Given that our primary objective was to assess the associations between variables, predictability was selected as an index to evaluate the significance of nodes in the generated network structure. For this reason, we only discussed predictability in this study and did not consider other parameters. Predictability measurements, which indicate the amount of variance explained by all other nodes in the network, quantify the extent to which a node is related by determining how well it can be predicted by all the other nodes in a network structure [31]. We computed percentages for the explained variance of continuous variables and for the correct classification of categorical variables to show how well a node could be predicted by its neighboring nodes on a scale of 0–1; 0 indicated that a node was not predicted by other nodes in the network, whereas 1 represented an ideal prediction.

In addition, the NetworkComparisonTest (NCT) package was used to evaluate network structure differences between male and female networks. A network structure invariance test was run to examine whether the global network structure differed between the two networks by comparing the distributions of edge weights. The global strength invariance test was used to compare the weighted absolute sum of edge weights of two networks [32,33].

## 3. Results

Table 2 presents the characteristics of the participants according to sex. The mean age of the participants was 34.76 ± 8.85. Of the 1276 participants, 60% were females and the majority were married. In terms of socioeconomic status, females tended to have a higher degree of education and economic condition than males. More females were married than males in the study.

The five components of well-being (positive emotion, engagement, relationships, meaning, and accomplishment) scored 20.18, 20.06, 20.75, 20.93, and 21.56, respectively, for the overall sample. Females were more likely to report better positive emotions and relationships than males. Engagement was reported to be marginally statistically significant (*p* = 0.053). There was no difference in accomplishment and meaning between male and female groups.

With respect to social participation, most individuals participated in housework, transportation, work, and physical activities (participation rate > 80%). Games, donation and volunteering activities had the least attendance (participation rate < 40%). Regarding sex differences in social participation, the participation rates in housework and physical activities were higher for females than males. Nevertheless, males had a higher participation rate in sports and games activities than females. There were no sex differences in the other types of social participation (Table 3).

Figure 1a presents the network structure estimated by the MGM in males; Table 4 shows the weight of each connection in the network. The results revealed a unique association between the variables suggesting that the economic condition was an important factor that affected individuals’ well-being, including positive emotion and engagement, and that was also related to housework and games activities. Age was also related to housework and games activities. Marital status was linked to sports and associations or social activities. Education was associated with housework, games, and work activities.

Among social participation and well-being, the edge between positive emotion–housework (0.263), positive emotion–games (0.102), engagement–housework (0.107), engagement–work (0.054), and meaning–socializing (0.085) had significant weights that showed a unique connection.

The predictability values of well-being were in the following order: meaning (variance = 0.814), accomplishment (variance = 0.805), positive emotion (variance = 0.765), relationships (variance = 0.600), and engagement (variance = 0.595), indicating that they were well predicted by other nodes in the network. In addition, the average predictability of all the nodes in the network was 0.417.

Figure 1b shows the network structure estimated by the MGM among females; Table 5 shows the weight of each connection in the network. The network structure suggested that the economic condition was related to relationships in well-being. Compared to males, socio-demographic variables were more strongly associated with the types of social participation in females.

Among social participation and well-being, the edges between positive emotion–physical activity (0.102), engagement–associations or societies (0.071), relationships–physical activity (0.090), relationships–socializing (0.092) and relationships–volunteering activities (0.133) had significant connections based on their weights.

The predictability values of well-being were in the following order: meaning (variance = 0.848), accomplishment (variance = 0.817), positive emotion (variance = 0.806), relationships (variance = 0.697), and engagement (variance = 0.659), indicating that they were well predicted by the other nodes in the network. The average predictability of all nodes in the network was 0.358.

The NCT analysis that compared the network between males and females showed no significant differences in network structure invariance (*p* = 0.194) and global strength invariance (*p* = 0.879). It followed that the overall structure of the two networks was similar but the internal associations were different.

## 4. Discussion

In this study, we found that social participation was beneficial to individual well-being. In addition, different social activities had different effects on well-being (i.e., positive emotion, engagement, relationships, meaning, and accomplishment) and there were differences according to sex between the two in the community population. Therefore, all three hypotheses of the study were confirmed. We first investigated the level of well-being and compared the differences based on sex. The scores for the five components of well-being in the overall sample were similar to the findings of a previous study among healthy adults [34]. Compared with males, females had higher positive emotion and relationship scores. These results also were reported in studies conducted in countries with both individualistic and collectivistic cultures [35,36]. Moreover, males experienced greater engagement than females. This may be because males scored higher in autonomy and purpose in life than females [23], so they would be more likely to have a higher engagement in daily life and work. Based on the results of this study, it is necessary to understand the differences in well-being between males and females.

In addition, we found differences according to sex in the types of participation in social activities. Compared with males, females have a higher proportion of participation in housework and physical activities and males were more likely to engage in sports and game activities than females. Intuitively, differences according to sex in social participation might nonetheless be expected given the existence of sex-differentiated physical fitness and social roles. In addition, interests and life experiences usually differ along certain lines for males and females [20]. These findings were in accordance with the results of other studies such as those by Li et al. and Xiao et al., which indicated that females tended to participate more in housework and light-intensity activities than males, who appeared to be more engaged outside the home, including participation in sports and socially oriented activities, in China [15,19]. These phenomena seem to hold true in both Eastern and Western cultures [20].

Furthermore, this study explored the effects of different types of social participation on well-being (i.e., positive emotion, engagement, relationships, meaning, and accomplishment) and examined the differences between them according to sex. Given that the complex associations among socio-demographic strata and different types of social participation might contribute to the various components of well-being, we adopted the MGM to understand the pairwise associations among them in both male and female samples. Although the overall structures of the two networks were not different, the internal associations between them were different. For males, after controlling for potential confounding factors such as age, marital status, education, and economic condition, housework was related to positive emotions and engagement. A plausible explanation for the results is that most of these activities are performed by females due to societal views and expectations [37]. Under such circumstances, males might view housework in a more relaxed way than females, who tend to view it as a task or job, so males are more likely to experience greater positive emotions and engagement during housework. Games, as a form of recreation, are generally relaxing [38], so the individual can experience more positive emotions. Socializing had a salubrious effect on well-being in this study, mainly for meaning. A study of older adults showed that engaging in social interactions was beneficial to well-being [39]. Being socially active is commonly encouraged by international guidelines for individual well-being [40]. One possibility is that the sense of belonging and relatedness that social interaction brings could make an individual feel greater meaning in life. Work as a productive and valuable social activity has a positive and significant effect on individuals’ well-being, especially engagement. These results were similar to those of other studies [41,42].

With respect to females, well-being was associated with physical, socializing, and volunteering activities. Research has documented that social activities are beneficial to well-being [14,43], but our findings clarified that they primarily influenced the good-relationship component of well-being. In addition, we found that physical activity was related to positive emotions. Greater engagement in well-being was reflected in the performance of activities in associations or societies. In the last few decades, there have been significant social changes for females (e.g., in employment and education). An increasing number of females are actively involved in associational or organizational activities and thus might have more positive experiences such as engagement. Sex partly reflects socially constructed norms, activities, behaviors, relationships, attributes, and opportunities that a given society considers appropriate for males and females [17]. As a result, males and females may differ in their well-being and social participation, as well as in their relationships. The differences, on the other hand, might also be due in part to the different benefits these activities bring to the male and female groups. Taken together, there are disparities in the relationship between social participation and well-being according to sex. From a public health perspective, these findings are important in identifying which types of social activity contribute to individual well-being; the effects of these social activities vary for males and females when considering the social roles.

This study had several limitations and the results should be interpreted with caution. First, the cross-sectional study design was limited in identifying the dynamic relationship between social participation and well-being. Second, the community samples were obtained from only one region of China that did not represent all Chinese individuals. In addition, according to the results, the average predictability of the network structure was 0.417 and 0.358 for the male and female groups, respectively. Therefore, some unmeasured confounders (such as nationality, type of work, gender, gender identity, and mental health) should be considered in future research. Third, although we considered various types of social participation, this range remained difficult to cover. In addition, we failed to consider the degree of social participation, willingness, and initiative to participate. Fourth, the data in this study were collected during the COVID-19 pandemic, which potentially affected people’s social activities and levels of well-being. This may have limited the representativeness of the results. Finally, the self-reported nature of the data might have led to bias to some extent.

## 5. Conclusions

In summary, males and females had different levels/types of well-being and social participation. There were also differences according to sex between social participation and well-being. Housework, games, and work activities were more strongly linked to well-being in males; for females, physical, socializing, and volunteering activities and memberships in associations or societies were related to greater levels of well-being. When considered from a public health perspective, this study offers a foundation for examining how specific activities are related to well-being in a community population and presents a more realistic description of the types of social activities in which they engage, which would most likely provide support to enhance public well-being. The government and media could actively publicize and encourage males and females to engage in different types of social activities. Therefore, it is important to identify the heterogeneity in the types of social activities according to sex. More studies in the future are required to identify the underlying mechanisms of sex in the relationship between social participation and well-being. Given the various types of social participation and components of well-being, the network analysis is a promising approach to exploring the complex associations between multiple variables.

## Figures and Tables

**Figure 1 ijerph-19-13135-f001:**
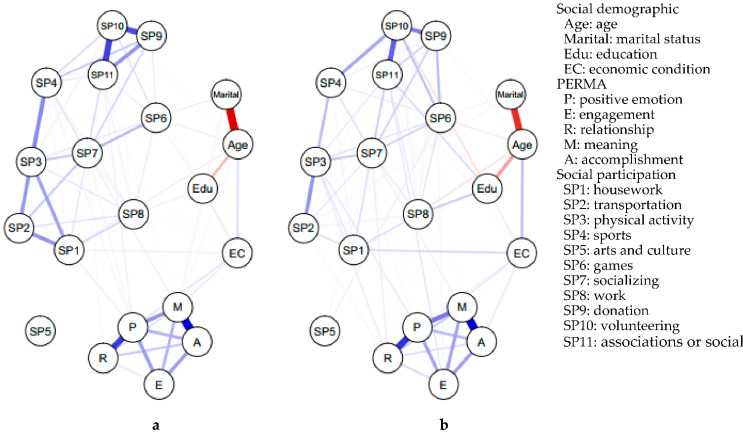
(**a**) Estimated network structure in male group (*n* = 510). (**b**) Estimated network structure in female group (*n* = 766). Circles represent variables and lines represent edges (i.e., association between two variables); the thicknesses and darkness of lines indicate the weights of the edges. Blue edges indicate positive association and red edges indicate negative association.

**Table 1 ijerph-19-13135-t001:** Information on social participation.

Category	Activity	Example
Activities of daily life	Housework	Cooking, cleaning, washing clothes, etc.
	Transportation	Bicycling, driving, using public transportation, etc.
Sports and entertainment activities	Physical activity	Walking, strolling, and other low-intensity physical activity
	Sports	Ball games, running, and other high-intensity physical activity
	Arts and culture	Reading, visiting concerts, museums, and art exhibits, etc.
	Games	Playing cards, games, chess, etc.
	Socializing	Seeing relatives, friends and neighbors; parties
Social service activities	Work	Full- or part-time job
	Donation	Donating money or goods to help others
	Volunteering	Voluntary work
	Associations or social	Mainly referred to activities outside of work

**Table 2 ijerph-19-13135-t002:** Descriptive statistics and *t*-test/chi-squared test results (*n* = 1276).

Variables	*N* (%)	Males (*n* = 510)	Females (*n* = 766)	*t*/*χ*^2^	*p*
Age (mean ± SD)	34.76 ± 8.85	33.57 ± 8.81	35.56 ± 8.79	3.968	<0.001
Marital status				26.774	<0.001
Married	914 (71.6)	324 (63.5)	590 (77.0)		
Other (single, widowed, and divorced)	362 (28.4)	186 (36.5)	176 (23.0)		
Education				55.377	<0.001
Senior high school or below	126 (9.9)	56 (11.0)	70 (9.1)		
College	218 (17.1)	134 (26.3)	84 (11.0)		
University or above	932 (73.0)	320 (62.7)	612 (79.9)		
Economic condition				39.387	<0.001
Insufficient	165 (12.9)	98 (19.2)	67 (8.7)		
Sufficient for essentials	714 (56.0)	289 (56.7)	425 (55.5)		
More than sufficient	397 (31.1)	123 (24.1)	274 (35.8)		
Well-being (mean ± SD)					
Positive emotion	20.75 ± 5.24	20.22 ± 5.05	21.10 ± 5.34	2.966	0.003
Engagement	20.06 ± 4.86	20.38 ± 4.69	19.85 ± 4.96	1.993	0.053
Relationships	21.56 ± 5.02	21.12 ± 4.89	21.85 ± 5.09	2.539	0.011
Meaning	20.93 ± 5.40	20.94 ± 5.19	20.93 ± 5.53	0.012	0.990
Accomplishment	20.18 ± 5.07	20.26 ± 4.89	20.12 ± 5.18	0.502	0.616

Note: SD, standard deviation.

**Table 3 ijerph-19-13135-t003:** Social participation data and differences based on sex (*n* = 1276).

Activity		*N* (%)	Males (*n* = 510)	Females (*n* = 766)	*χ* ^2^	*p*
Housework	No	71 (5.6)	47 (9.2)	24 (3.1)	20.414	<0.001
	Yes	1205 (94.4)	463 (90.8)	742 (96.9)		
Transportation	No	102 (8.0)	50 (9.8)	52 (6.8)	3.386	0.066
	Yes	1174 (92.0)	460 (90.2)	714 (93.2)		
Physical activity	No	202 (15.8)	98 (19.2)	104 (13.6)	6.888	0.009
	Yes	1074 (84.2)	412 (80.8)	662 (86.4)		
Sports	No	469 (36.8)	156 (30.6)	313 (40.9)	13.462	<0.001
	Yes	807 (63.2)	354 (69.4)	453 (59.1)		
Arts and culture	No	454 (35.6)	185 (36.3)	269 (35.1)	0.132	0.716
	Yes	822 (64.4)	325 (63.7)	497 (64.9)		
Games	No	768 (60.2)	229 (44.9)	539 (70.4)	81.785	<0.001
	Yes	508 (39.8)	281 (55.1)	227 (29.6)		
Socializing	No	381 (29.9)	149 (29.2)	232 (30.3)	0.121	0.728
	Yes	895 (70.1)	361 (70.8)	534 (69.7)		
Work	No	194 (15.2)	71 (13.9)	123 (16.1)	0.924	0.336
	Yes	1082 (84.8)	439 (86.1)	643 (83.9)		
Donation	No	823 (64.5)	325 (63.7)	498 (65.0)	0.169	0.681
	Yes	453 (35.5)	185 (36.3)	268 (35.0)		
Volunteering	No	927 (72.6)	360 (70.6)	567 (74.0)	1.647	0.199
	Yes	349 (27.4)	150 (29.4)	199 (26.0)		
Associations or social	No	747 (58.5)	311 (61.0)	436 (56.9)	1.917	0.166
	Yes	529 (41.5)	199 (39.0)	330 (43.1)		

**Table 4 ijerph-19-13135-t004:** The weight of each connection in the network structure (males).

Variables	Age	Marital Status	Education	Economic Condition	Positive Emotion	Engagement	Relationships	Meaning	Accomplishment
Positive emotion	0	0	0	0.065					
Engagement	0	0	0	0.047					
Relationships	0	0	0	0					
Meaning	0	0	0	0					
Accomplishment	0	0	0	0					
Housework	0.205	0	0.088	0.124	0.263	0.107	0	0	0
Transportation	0	0	0	0	0	0	0	0	0
Physical activity	0	0	0	0	0	0	0	0	0
Sports	0	0.162	0	0	0	0	0	0	0
Arts and culture	0	0	0	0	0	0	0	0	0
Games	0.201	0	0.148	0.045	0.102	0	0	0	0
Socializing	0	0	0	0	0	0	0	0.085	0
Work	0	0	0.165	0	0	0.054	0	0	0
Donation	0	0	0	0	0	0	0	0	0
Volunteering	0	0	0	0	0	0	0	0	0
Associations or social	0	0.236	0	0	0	0	0	0	0

**Table 5 ijerph-19-13135-t005:** The weight of each connection in the network structure (females).

Variables	Age	Marital Status	Education	Economic Condition	Positive Emotion	Engagement	Relationships	Meaning	Accomplishment
Positive emotion	0	0	0	0					
Engagement	0	0	0	0					
Relationships	0	0	0	0.085					
Meaning	0	0	0	0					
Accomplishment	0	0	0	0					
Housework	0	0	0	0.363	0	0	0	0	0
Transportation	0	0.280	0.128	0	0	0	0	0	0
Physical activity	0	0.135	0.118	0	0.102	0	0.090	0	0
Sports	0.129	0	0.073	0	0	0	0	0	0
Arts and culture	0	0	0.101	0	0	0	0	0	0
Games	0.142	0	0.095	0	0	0	0	0	0
Socializing	0	0	0	0.047	0	0	0.092	0	0
Work	0.190	0	0.201	0.066	0	0	0	0	0
Donation	0	0	0	0	0	0	0	0	0
Volunteering	0	0	0.420	0.356	0	0	0.133	0	0
Associations or social	0	0	0.254	0.049	0	0.071	0	0	0

## Data Availability

The data presented in this study are available upon request from the corresponding author.

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
