# Peer review of "Differences According to Sex in the Relationship between Social Participation and Well-Being: A Network Analysis"

_ijerph, 2022, doi:10.3390/ijerph192013135_

Round 1

Reviewer 1 Report

The problem of well-being is very important in this times. A analysis of relationship between social participation and well-being is even more important in terms of post COVID-19 society. Therefore the study conducted by the Authors has a practical dimension and is very important and up to date in many contexts, e.g.: social, demographic, economic, administrative, spatial.

The literature review basically does not exist. A few of the references provided are not exhaustive. It definitely needs supplementing.

Most of the literature items presented in the reference list are current. They are all related to the topic presented.

However, I think that on such a vast subject, using 34 references only is not enough. The analysis of the literature should be deepened. The Authors focused very much on the methodological part and on the analysis of the data and their interpretation, but the topic itself requires deepening and supplementing.

  The title is adequate to the research problem being undertaken. The paper has been correctly divided into relevant sections, and their content coincides with their titles. However, a literature review section should be added.

The introduction part is interesting, but unfortunately it does not show how the studies phenomenon is presented in the subjects literature. Only a few publications on this topic were cited. In my opinion, it is not enough.

Footnotes and bibliography are in my opinion correctly formulated.

The correct terminology was used. The language of the article is correct, adequate.   Research hypotheses was given. Authors should refer to them at the end of the paper.

Conclusions part should be extended.

Reviewer 2 Report

This study aimed to explore the effects of different types of social participation on the components of well-being, as well as the gender differences in the relationship between social participation and well-being. Therefore, the use of the statistical method "Social network" is a new addition because the idea of research is closer to the traditional.

Too many social activities may cause confusion, so if it is possible to shorten it to the large category that includes these activities, it will be better.

Reviewer 3 Report

The manuscript reports an interesting study about the role of social participation in people's well-being, looking for differences linked to gender. The manuscript is well written, and the results are reported clearly. I have only a few comments for the authors that want to be helpful for the improvement of the manuscript. 

- Looking at the demographic description: were all the participants cisgender? Are you sure that you have to evaluate gender and not sex? Have you evaluated sexual orientations? These are all elements that have a role in well-being and social participation as well.

- How did you define poor/fair/good economic conditions?

- Might the differences between genders have a role in the results? Is there any justification in your opinion for these differences?

- There is a specific package in R called NetworkComparisonTest that can be used to evaluate differences between networks. Have you evaluated the possibility of performing this analysis?

- You stated that people with several physical conditions were excluded; how did you evaluate this aspect? What about mental health?

- Your figures are not clear about the edges. Could you please consider evaluating a change of colors? Because it is very difficult to evaluate the positive or the negative connections. 

Round 2

Reviewer 3 Report

I think the authors have addressed quite all my concerns. I have only one comment that needs to be considered:

- In the first round I asked about the evaluation of gender or sex, and the authors declared that they did not evaluate gender identity. Thus, I think the manuscript should be revised by changing all the "gender" into "sex" because they evaluate that aspect. Indeed, gender refers to the characteristics of women, men, girls, and boys that are socially constructed, while sex refers to the physical differences between people who are male, female, or intersex.
